# Changes in the Serum Concentration Levels of Serotonin, Tryptophan and Cortisol among Stress-Resilient and Stress-Susceptible Individuals after Experiencing Traumatic Stress

**DOI:** 10.3390/ijerph192416517

**Published:** 2022-12-08

**Authors:** Ewa Alicja Ogłodek

**Affiliations:** Department of Health Sciences, Jan Dlugosz University, 42-200 Częstochowa, Poland; e.oglodek@ujd.edu.pl

**Keywords:** serotonin, stress-resilient, stress-susceptible, traumatic stress, tryptophan

## Abstract

Background: Stress is a common response to many environmental adversities. However, once dysregulated, this reaction can lead to psychiatric illnesses, such as post-traumatic stress disorder (PTSD). Individuals can develop PTSD after exposure to traumatic events, severely affecting their quality of life. Nevertheless, not all individuals exposed to stress will develop psychiatric disorders, provided they show enhanced stress-resilience mechanisms that enable them to successfully adapt to stressful situations and thus avoid developing a persistent psychopathology. Methods: The study involved 93 participants. Of them, 62 comprised a study group and 31 comprised a control group. The aim of the study was to assess serotonin, cortisol and tryptophan concentration levels in subjects with PTSD (stress-susceptible; PTSD-SS) and in healthy individuals (stress-resilient; PTSD-SR), who had experienced a traumatic event but fully recovered after the trauma. The subjects were between 18 and 50 years of age (mean 35.56 ± 8.26 years). The serum concentration levels of serotonin, cortisol and tryptophan were measured with an ELISA kit. Results: It was found that the serotonin, tryptophan and cortisol concentration levels were consistent with the features of both PTSD-SR and PTSD-SS patients. It was reported that the mean cortisol concentration levels increased more significantly in the PTSD-SS group than in the PTSD-SR group, versus those in the control group. Similarly, the PTSD-SS group was found to show a larger decrease in the mean serotonin concentration levels than the PTSD-SR group, versus those in the control group. No significant changes were found in the tryptophan concentration levels between the study groups, versus those in the control group. Conclusions: These findings can be useful when attempting to improve resilience in individuals using neuropharmacological methods. However, it is necessary to conduct more cross-sectional studies that would address different types of negative stress to find out whether they share common pathways.

## 1. Introduction

Post-traumatic stress disorder (PTSD) can be diagnosed based on the Diagnostic and Statistical Manual of Mental Disorders (DSM-5) criteria. They include direct or indirect exposure to a traumatic event, which is followed by symptoms divided into four categories, namely intrusion, avoidance, negative changes in thoughts and mood and alterations in arousal and reactivity. Exposure to traumatic events can trigger PTSD and severely affect the quality of life [1,2,3]. The disorder leads to certain physiological symptoms, such as elevated arousal and reactivity, e.g., anger out-burst, irritability, reckless behavior with no concern for consequences, hypervigilance, sleep disturbance and limited focus [4,5,6]. The pathogenesis of PTSD is related to the maladaptive expression of traumatic memories, which is characterized by excessive consolidation, generalization, and fear extinction deficits [7]. It is interesting to note that not every person who experiences trauma will develop PTSD. According to many studies [8,9,10,11], exposure to trauma in the general population is high (about 90%), while the prevalence of PTSD amounts to around 7–8%. Other authors [12,13,14,15] have cited evidence indicating an increased risk of PTSD in individuals whose parents have experienced trauma and/or suffered from PTSD. Increasingly, scientific studies have been addressing the subject of susceptibility to stress and have distinguished trauma-susceptible and trauma-resilient individuals [16,17]. Some of the criteria used to evaluate vulnerability to stress include anxiety and the neurohormonal responses in the brain. Stress stimulates the hypothalamic–pituitary–adrenal (HPA) axis, which coordinates an inflammatory reaction through a neurohormonal cascade. Cortisol activates the glucocorticoid receptors (GRs)—distributed in the brain, mainly along the HPA axis and in the limbic structures, such as the amygdala body complex (AMY), the hippocampus (HIP) and the prefrontal cortex (PFC) [18,19,20]. Trauma-resilience refers to the overcoming of traumatic stress. Such individuals show reduced vulnerability to environmental risk experiences and can maintain relatively normal physical and psychological function. By contrast, trauma-susceptibility can be referred to as the detrimental physiological and psychological consequences of trauma exposure, including serious stress-related psychiatric conditions (e.g., PTSD). The neurobiological features of resilience have been discussed in the literature. Individuals who are able to adapt to traumatic stress may show changes in biological processes that differ from those found in stress-susceptible individuals. The biological basis of resilience to stress is represented by many biomarkers, for example, cortisol, serotonin and tryptophane, which are usually examined separately [19,20].

Cortisol has been shown to have numerous effects on the body, such as increasing the sensitivity of the thalamus to incoming stimuli. Being persistently exposed to high levels of cortisol can make the nervous system sensitized to stimuli of a psychologically threatening nature, the so called kindling concept. In terms of chronic trauma, a stronger psychological and physiological reaction is induced by trauma experiences of diminishing strength. Individuals who experience trauma are oftentimes observed to have increased cortisol reactivity. Moreover, elevated levels of cortisol have been shown to improve the recall of emotionally significant information and consolidate long-term memory. As a result, improved memory for traumatic events can impact the evaluation of similar events in the future. This may lead to the intrusive recollection of the trauma through images, thoughts or perceptions [8,16,17,18,19,20].

The hippocampus and the medial prefrontal cortex are modulated by serotonin (5-HT) via receptors [21]. It is worth noting that the 5-HT fibers rarely have direct synaptic contacts. Moreover, the 5-HT receptors are located on neurons that are not serotonergically innervated. This shows that in the hippocampus, as in other areas of the brain, 5-HT is diffusely released through volume transmission and serves as a neuromodulator to maintain homeostasis in the brain [22,23,24,25].

Serotonin has also been shown to play a role in traumatic stress. In the case of moderate stress, serotonin is released into the frontal cortex to calm and diminish dysphoria and anxiety symptoms. As for severe stress or trauma, excessive serotonin activation is observed in many regions of the brain. High serotonin levels can lead to serotonin depletion if the trauma is chronic or persistent [21,22].

Reexperiencing the trauma and intrusive memories, even if no trauma is happening, causes chronic serotonin activation. This inhibits the CNS to dampen emotional reactions to later stressors. Serotonin depletion may also trigger hyperarousal observed in PTSD, including hypervigilance, impulsivity and irritability [24,25].

Serotonin is a monoamine neurotransmitter synthesized from tryptophan—an amino acid. Neurons that contain 5-HT are derived from the dorsal and medial raphe nuclei in the brainstem and target multiple areas of the forebrain, including the amygdala, the bed nucleus of the stria terminalis, the hippocampus, the hypothalamus and the prefrontal cortex [26,27,28]. 5-HT is responsible for regulating sleep, appetite, sexual behavior, aggression/impulsivity, motor function, analgesia and neuroendocrine function. Through its involvement in the neurohormonal homeostasis of the brain, 5-HT controls stress responses during traumatic experiences [29,30,31]. Research shows that the 5-HT neurons of the dorsal raphe mediate anxiety effects via the 5-HT2 receptors in the amygdala and hippocampus [32,33,34]. By contrast, the 5-HT neurons of the middle raphe are said to mediate anti-anxiety effects, facilitate extinction and suppress the encoding of learned associations via the 5-HT1A receptors. Based on animal models, it has been found that chronic exposure to stressors induces the upregulation of 5-HT2 and the downregulation of 5-HT1A receptors [35,36,37]. Serotonin deficiency is associated with the development of PTSD symptoms, such as impulsivity, hostility, aggression, depression and suicidal thoughts [38]. Other findings of altered 5-HT neurotransmission in PTSD include decreased serum 5-HT concentrations, decreased density of 5-HT reuptake sites in platelets and an altered response to CNS serotonergic provocation [39,40]. To sum up, altered 5-HT transmission may contribute to the development of PTSD symptoms, such as hypervigilance, increased surprise, impulsivity and intrusive memories.

The aim of this study was to evaluate changes in the concentration of serotonin, tryptophan and cortisol in PTSD patients from both stress-resilient and stress-susceptible groups.

## 2. Materials and Methods

### 2.1. Study Design and Participants

Prior to being enrolled in the study, all participants had signed an informed patient consent form. The study comprised 93 males between 18 and 50 years of age (mean 35.56 ± 8.26 years). Both the study and control groups involved rescuers from the Central Mine Rescue Station, mine rescue workers and firefighters. The recruitment process was conducted at the Medical Center. The groups consisted of 31 controls, 32 subjects with PTSD (stress-susceptible) and 30 healthy subjects who had experienced a traumatic event similar to that of PTSD patients but fully recovered after the trauma. The characteristics of the study groups are shown in Table 1. The exclusion criteria for the study groups involved the occurrence of the following clinical situations in the subjects (after enrollment in the study): mental illness other than PTSD, central nervous system damage, alcohol or other substance abuse, chronic illness and infectious disease, medication use, and heavy smoking. None of the patients enrolled in the study had to be excluded. The inclusion criteria for the controls were as follows: no psychiatric or somatic illnesses, no prior use of psychiatric medications, no history of traumatic incidents—up to 2 in a lifetime, the number ascending relatives—up to 1 person in the family, and no heavy smoking.

The fifth edition of the DSM-5 was adopted for enrollment [41]. The examination was conducted by the first author of this paper—a specialist in psychiatry and family medicine.

### 2.2. Serum Analytical Procedures

The analytical procedures took place between 8:00 and 10:00 in the morning to evaluate serotonin, tryptophan and cortisol concentration levels. Fasting blood samples (15 mL) were collected into sterile chilled tubes using a standard venipuncture technique. The samples were left to clot at room temperature and were then centrifuged at 3500 rpm for 10 min at 4 °C. The samples were stored at −80 °C until lab analysis. The concentration levels of serotonin were measured using an ELISA kit from Labor Diagnostika NORD, Nordhorn, Germany; those of cortisol were measured using an ELISA kit from Abbott, U.S.; and those of tryptophan were analyzed using an ELISA kit from Labor Diagnostika NORD, Nordhorn, Germany. The measurements were performed according to the manufacturers’ instructions and calculated by means of a standard straight line.

### 2.3. Statistical Analysis

The statistical calculations were performed using the software environment R v.4.1.1 and IDE RStudio v. 1.4.1717. The analysis was based on the built-in functions of the following packages: “psych” (descriptive statistics), “tidyverse”, “dplyr” (manipulations on data frames), “rstatix” (statistical tests) and “ggstatsplot” (graphical reporting). The significance level in this study was α = 0.05. To measure the distribution of variables, mean, standard deviation, median, and minimum and maximum values, the skewness and kurtosis measures were calculated. To test the research hypothesis, each sample was analyzed by means of the so-called one-way Welch’s ANOVA followed by non-parametric post-hoc tests, such as Games-Howell’s analysis with the partial omega-squared effect size [42,43,44,45,46].

### 2.4. Ethics

The study was carried out in accordance with the Declaration of Helsinki of the World Medical Association. Its protocol was approved by the Ethics Committee (No. 39/2018). All subjects provided written informed consent to their examiner and at the same time, to the author of this article. The author of this paper is also the leader of the Polish Team for the project entitled: ‘Mapping and Interrogating Top-Down Control of the Memory Engram of the Posttraumatic Stress Disorder’, under the ERA-NET NEURON COFUND grant, NCBiR 19/2020, under which this study was conducted.

## 3. Results

This study assessed the concentration levels of serotonin, tryptophan and cortisol in both the PTSD-SS (PTSD—stress-susceptible) and PTSD-SR (PTSD—stress-resilient) groups. A 2.5-fold decrease in the serotonin levels was reported for the PTSD-SS group versus those in the control group and a 1.7-fold decrease was reported for the PTSD-SR group versus those in the control group. Welch’s ANOVA analysis of the mean measures for serotonin showed a statistically significant main effect of *F* (2; 51.42) = 89.34, *p* < 0.001, with an effect size of $w_{p}^{2}$ = 0.76, indicating significant differences between the group means. The Games-Howell post hoc test with a Bonferroni adjustment indicated significant differences between the group pairs: control group (M = 260.45; SD = 61.0)—PTSD-SR group (M = 106.90; SD = 26.41); PTSD-SS group (M = 106.90; SD = 26.41)—PTSD-SR group (M = 150.05; SD = 14.8); control group (M = 260.45; SD = 61.0)—PTSD-SR group (M = 150.05; SD = 14.8). The results of the analysis are shown in Figure 1 and Table 2.

The next analyzed parameter was tryptophan. No statistically significant differences were found between the groups: control group (M = 10.94; SD = 3.75)—PTSD-SR group (M = 11.55; SD = 4.34); PTSD-SS group (M = 14.37; SD = 27.65)—PTSD-SR group (M = 11.55; SD = 4.34); control group (M = 10.94; SD = 3.75)—PTSD-SR group (M = 11.55; SD = 4.34). The results are shown in Figure 2.

Welch’s ANOVA analysis of the mean measures showed no statistically significant main effect *F* (2; 57.11) = 2.76, *p* = 0.070, i.e., no significant differences in the tryptophan concentration levels between the groups. The effect size was found to be moderate, $w_{p}^{2}$ = 0.06, indicating that the effect could have been significant if the sample size had been increased. The results of the analysis are shown in Figure 2 and Table 2.

Another parameter assessed was cortisol. Here, an increase in the cortisol levels was reported for the PTSD-SS group compared to those of the control group. A slight increase was also shown for the PTSD-SR group versus the control group. The mean measures between the rest of the pairs were found to be statistically insignificant. The results are shown in Figure 3.

Welch’s ANOVA analysis of the mean measures revealed a statistically significant main effect of F (2; 60.00) = 6.07, p = 0.004, with an effect size of $w_{p}^{2}$ = 0.14, indicating significant differences between the group means.

The Games-Howell post hoc test with a Bonferroni adjustment indicated significant differences between the following groups: control group (M = 10.47; SD = 2.97)—PTSD-SS group (M = 13.09; SD = 3.07).

Based on the data presented in Table 2, it was shown that for most of the variables, the results of the Shapiro-Wilk test indicated a distribution different from normal (*p* < 0.05). However, important factors showing the type of distributions of variables are the values of statistics characterizing the shapes and symmetries of a given distribution, namely skewness and kurtosis. In this regard, most of the variables were characterized based on a measure of skewness with an absolute value of less than 2.0 and kurtosis less than 7.0. Based on the results of Curran P.J. et al. [47], Bandalos D.L. [43], and Fabrigar L.R. et al. [48], unimodal distributions with values of skewness less than 2.0 and kurtosis less than 7.0 can be treated as variables with normal distributions. Therefore, in order to test the significance of differences between the groups for serotonin, tryptophan and cortisol concentrations, parametric tests were used.

## 4. Discussion

Post-traumatic stress disorder is a severely debilitating condition experienced by a multitude of individuals. Its symptoms, if left untreated, cause a number of adverse health outcomes [49].

Individuals experiencing traumatic stress are observed to suffer from neurobiological abnormalities. This indicates the dysregulation of multiple biological systems involved in the body’s protection against stress [50,51]. Apparently, this pathophysiological dysregulation occurs in individuals with genetic, epigenetic and psychological predispositions exposed to trauma-related experiences [4,52].

Many studies [53,54,55,56,57] show that early adverse life experiences have long-lasting effects on the neurobiological system functioning. They claim that these experiences somehow “program” subsequent stress reactivity and vulnerability to PTSD. Individuals predisposed to PTSD are characterized by their inability to process traumatic experiences without leaving traces of sensorimotor dysfunction. The body becomes sensitized to the experienced trauma, which leads to impaired homeostasis in the body. Moreover, the literature reports that the risk of PTSD becomes increased if a parent suffers from PTSD due to either previous traumatic events or the lack of environmental support [58]. Clinically, there is an increased risk of PTSD in individuals classified as “susceptible to stress” compared to a lower risk of PTSD in individuals who experienced trauma but did not develop PTSD—referred to as “resilient to stress”. In this context, individuals classified as “stress resilient” are those who demonstrate an increased ability to avoid the harmful physiological and psychological consequences of stress resulting in the development of various post-traumatic psychiatric disorders, such as major depression (MDD) or anxiety disorders [59].

This study involved miners, mine rescuers and firefighters. In the present study, the serum concentration levels of cortisol, serotonin and tryptophan were compared among the following groups of subjects: PTSD-SS, PTSD-SR, and controls.

Neurohormonal dysregulation observed in PTSD is related to HPA axis dysregulation and stress vulnerability [49]. So far, neurohormonal post-stress vulnerability studies have been performed using animal models [14,60,61]. Torrisi S.A. et al. [60] found significantly higher basal post-trauma corticosterone levels only in susceptible mice when compared to both pre-trauma basal corticosterone levels and post-trauma basal corticosterone levels in control and resistant mice.

This study can be considered innovative as it determined changes in serotonin, tryptophan and cortisol among PTSD patients from both stress-resilient and stress-susceptible groups. PTSD-SS patients experienced more traumatic incidents and were more burdened by the traumatic experience due to their ascending relatives (also working in hazardous conditions) than the PTSD-SR and control groups.

Elzinga B.M. et al. [62] assessed cortisol responsivity to traumatic recollections among males with childhood abuse-related PTSD. PTSD patients were found to have higher cortisol responses to stress in comparison to those of controls.

D’Elia A.T.D. et al. [63] reported increased salivary cortisol levels in women with sexual abuse-related PTSD. Maes et al. [64] also reported increased urinary cortisol levels in a group of survivors of two traumatic events that occurred within just a few years of the published report: a flash fire and a multiple-vehicle collision. Field, T. et al. [65] found that decreased cortisol levels and increased serotonin and dopamine levels were positive effects of massage therapy. Similarly, elevated cortisol levels were observed by Song Y. et al. in survivors of earthquake trauma [66].

Morris M.C. et al. [67] pointed out that hypercortisolemia during the course of PTSD affects the function of the HPA axis and thus leads to the development of depressive disorders after post-traumatic stress. Post-traumatic stress activates the HPA axis, adversely affecting not only the brain, but also the immune system, which is the body’s first line of defense. The literature suggests that chronic stress promotes the persistence of inflammation [68,69]. Pro-inflammatory signals can directly reach the brain or communicate with microglia through the neurovascular network. The inflammatory response in the renal system persists and negatively impacts the neurohormonal system of the brain [70].

It is important to note that chronic stress in PTSD can lead to depression. PTSD and depression often show common biological links related to sensitization of the central nervous system [23,25]. This affects such centers as the amygdala, the anterior cingulate cortex and the periaqueductal gray matter. At the same time, these areas belong to the limbic system, responsible for regulating emotions. Moreover, PTSD is associated with increased cortisol secretion and impaired serotonin and tryptophan secretion [27,35].

Hypercortisolemia may decrease the availability of tryptophan (TRP), being a substrate for the synthesis of serotonin (5-HT).

TRP is the precursor of 5-HT and is broken down in the diet by the predominant kynurenine (KYN) pathway. The KYN pathway is closely related to immunity. The literature shows that both stress and inflammation increase KYN production by activating the TRP-degrading enzymes, diverted from 5-HT synthesis [71].

The increased production of KYN may be involved in the pathogenesis of mood disorders because overactivity of the KYN pathway triggers the inflammatory process with stress. It has been shown that under the influence of excess cortisol, the hippocampus may undergo atrophy. Other authors have also reported decreased cortisol levels in PTSD. Low cortisol levels can also be devastating to the central nervous system. Cortisone deficiency may lead to insufficient glucose utilization by neurons and inhibition of the synthesis of neurotransmitters. This may also lead to the development of depression [72,73].

To sum up, it should be noted that increased cortisol levels in combination with decreased serotonin levels and abnormal regulation of the HPA axis may affect the nervous system function and become a predictor of other psychiatric disorders.

Moreover, recurring PTSD symptoms can persist for many years, and in acute cases—even for a lifetime. Moreover, PTSD symptoms can resemble depression. Typical PTSD symptoms include excessive fatigue (neurasthenia) after physical or mental exertion and nightmares, in which one returns with thoughts of the traumatic event. As a result, insomnia can occur. In addition, PTSD-affected individuals often experience anxiety attacks and are emotionally unstable. As a result of strong experiences, there is also a risk of suicide in the moment of such an attack [23,25]. Therefore, people with PTSD, especially those who belong to the PTSD-SR group, should have easier access to psychological and psychiatric assistance.

## 5. Conclusions

The above findings show that long-term stress, such as PTSD, is associated with the upregulation of inflammatory agents. Chronic stress may elevate stress perception and thus activate the production of pro-inflammatory messenger molecules responsible for the development of depression.

The novelty of this study is that it highlights the existence of the phenomenon associated with increased sensitivity to trauma in individuals who experience PTSD and often have a family history of trauma in the second and/or third generation. This is reflected by greater changes in serotonin and cortisol levels compared to those in PTSD-SR individuals. Therefore, PTSD-SS individuals should be provided with mental health preventive measures, as their risk of mental illness is higher than that of trauma-resilient individuals (PTSD-SR). These results can contribute to future research on anti-inflammatory markers.

## Figures and Tables

**Figure 1 ijerph-19-16517-f001:**
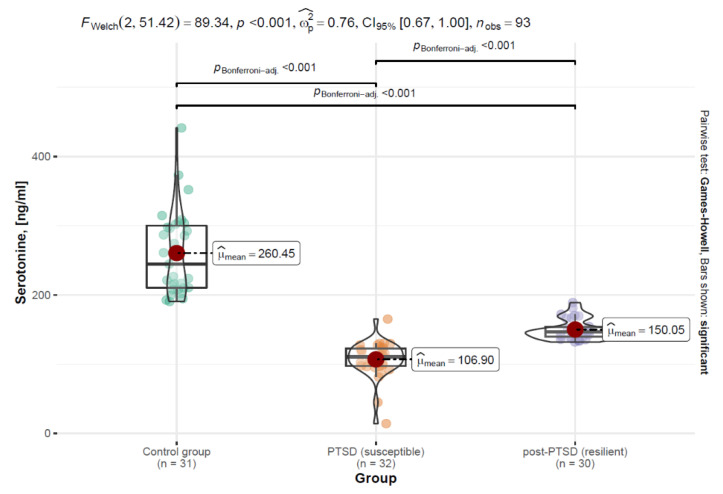
Distribution of the serotonin variable by patient groups in the form of a violin plot, including statistical tests and differences between the groups.

**Figure 2 ijerph-19-16517-f002:**
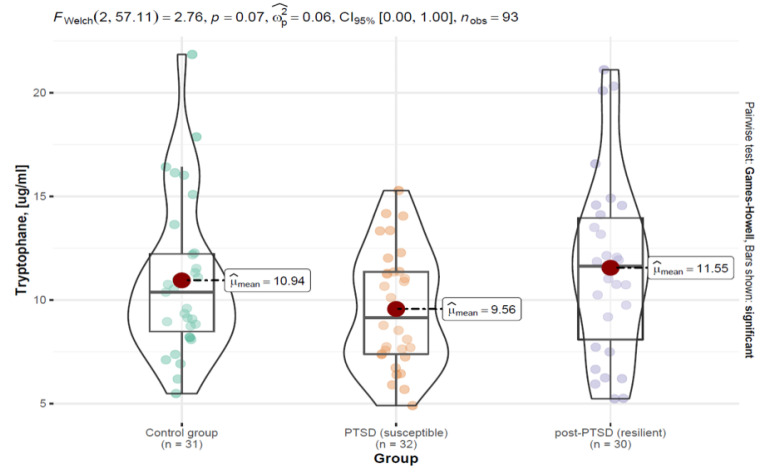
Distribution of the tryptophan variable by patient groups in the form of a violin plot, including statistical tests and differences between the groups.

**Figure 3 ijerph-19-16517-f003:**
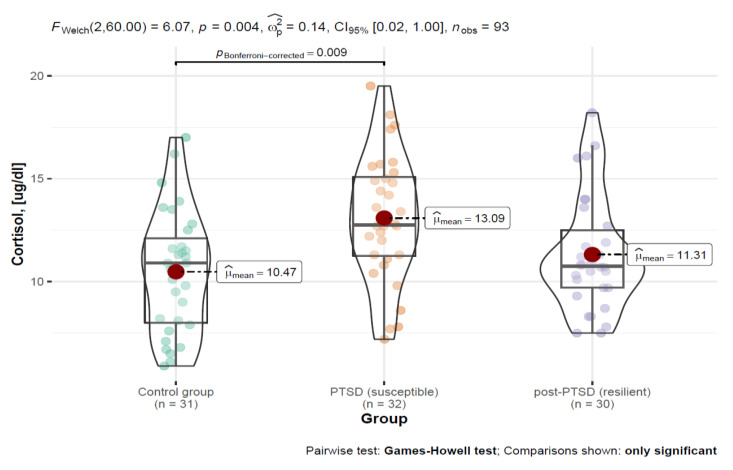
Distribution of the cortisol variable by patient groups in the form of a violin plot, including statistical tests and differences between the groups.

**Table 1 ijerph-19-16517-t001:** Characteristics of study groups.

Study Groups	*n*	M	SD	*p* C vs. SR	*p* C vs. SS	*p* SR vs. SS	
Age of participants							
Control	31	35.41	+/− 10.18	0.844	0.969	0.770	
PTSD-SR	32	35.88	+/− 7.56				
PTSD-SS	30	35.35	+/− 7.05				
Years of Working in Hazardous Conditions							
Control	31	11.59	+/− 9.41	0.074	0.074	0.714	
PTSD-SR	32	16.73	+/− 8.50				
PTSD-SS	30	16.03	+/− 6.90				
Number of Traumatic Incidents Experienced at Work							
Control	31	1.44	+/− 1.52	<0.001	<0.001	0.249	
PTSD-SR	32	23.82	+/− 16.24				
PTSD-SS	30	27.94	+/− 12.46				
Number of Ascending Relatives Working in Hazardous Conditions							
Control	31	0.38	+/− 0.49	<0.001	<0.001	0.003	
PTSD-SR	32	3.06	+/− 0.97				
PTSD-SS	30	2.24	+/− 1.02				

Legend: Control—control group, PTSD-SR—PTSD stress-resilient group; PTSD-SS—PTSD stress-susceptible group; *n*—group sample; *M*—mean; *SD*—standard deviation, *p* C vs. SR—adjusted *p*-value between Control and PTSD-SR, *p* C vs. SS—adjusted *p*-value between Control and PTSD-SS, *p* SR vs. SS—adjusted *p*-value between PTSD-SR and PTSD-SS.

**Table 2 ijerph-19-16517-t002:** Descriptive statistics of the variables with the use of ratio and interval scales by study groups.

CONTROL GROUP								
	*n*	M	SD	Mdn	Min.	Max.	Sk.	Kurt.
serotonin	31	260.45	61.02	244.50	190.50	441.30	0.92	0.41
tryptophan	31	10.94	3.75	10.37	5.48	21.85	0.98	0.49
cortisol	31	10.47	2.97	10.90	5.90	17.00	0.27	−0.79
PTSD (SUSCEPTIBLE) GROUP								
serotonin	32	106.90	26.41	110.55	14.10	165.10	−1.35	3.60
tryptophan	32	9.56	2.81	9.14	4.90	15.28	0.26	−1.13
cortisol	32	13.09	3.07	12.75	7.20	19.50	−0.04	−0.62
POST-PTSD (RESILIENT) GROUP								
serotonin	30	150.05	14.84	146.80	132.00	188.90	0.98	−0.02
tryptophan	30	11.55	4.34	11.62	5.23	21.11	0.48	−0.41
cortisol	30	11.31	2.77	10.75	7.50	18.20	0.77	−0.19

Legend: *n*—group sample; M—mean; SD—standard deviation; Mdn—median; Min and Max.—minimum and maximum value of the distribution; Sk.—skewness; Kurt.—kurtosis.

## Data Availability

The datasets used and analyzed during the current study are available from the corresponding author on reasonable request.

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
