# Peer review of "Changes in the Serum Concentration Levels of Serotonin, Tryptophan and Cortisol among Stress-Resilient and Stress-Susceptible Individuals after Experiencing Traumatic Stress"

_ijerph, 2022, doi:10.3390/ijerph192416517_

Round 1

Reviewer 1 Report

Congratulations for the article!

I think that the resilience must be discussed in the introduction and how it was measured. 

The references have 40% out dated (prior to 2017. So, I strongly recommend to update til 80%.

Author Response

Dear Editor,

In response to the review of the paper titled: ‘Changes in the Serum Concentration Levels of Serotonin,
Tryptophan and Cortisol among Stress-Resilient and
Stress-Susceptible Individuals after Experiencing Traumatic Stress’ (ijerph-2005992) the relevant changes suggested by the editor have been made.

In accordance with the recommendations of Reviewer 1:

    1. Resilience of PTSD-SR and PTSD-SS has been described.
    2. The indicated publications have been updated.

Kindest regards

The Author

Reviewer 2 Report

The main question is “Is there any difference in serotonin, tryptophan and cortisol concentration between stress‐resilient and stress‐susceptible PTSD patients. Concentration of the hormones in PTSD is definite but in this study their concentrations have between stress‐resilient and stress‐susceptible PTSD patients, which is original and novel. The findings of this study can be helpful in treatment of PTSD patient who are stress‐resilient or stress‐susceptible. Also, based on the results of this study, any preventive actions for PTSD can be done in stress‐susceptible. Effects of other diseases on the difference should be studied. Maybe the patients have some diseases which can change the results.

1.       Exclusion criteria are reasons that subjects who included in the study as samples, will be omitted omitted from the study and she/he cannot be permitted to continue the study not criteria of not inclusion. Therefore, the mentioned content in lines 92-94 is not exclusion criteria.

2.       Please statistically compare the variables in table 1 between 3 groups and mentioned related P value.

3.       P values in table 2 are related to comparison between which groups??

4.   The references are good but better to add some similar studies.

Author Response

Dear Editor,

In response to the review of the paper titled: ‘Changes in the Serum Concentration Levels of Serotonin,
Tryptophan and Cortisol among Stress-Resilient and
Stress-Susceptible Individuals after Experiencing Traumatic Stress’ (ijerph-2005992) the relevant changes suggested by the editor have been made.

In accordance with the recommendations of Reviewer 2:

    1. The exclusion criteria have been corrected, lines 92-94
    2. The references have been supplemented.
    3. Re p.2 of the review:  The results of the significance of the group comparisons performed have been added in Table 1.
    4. Re p.3 of the review:  These were the results of the Shapiro-Wilk test. They have been omitted from the revised version so as not to cause confusion.

Kindest regards

The Author

Reviewer 3 Report

It is an interesting work. Authors have evaluated the changes in serotonin, tryptophan and cortisol levels in PTSD patients from both stress‐resilient and stress‐susceptible groups. The manuscript is well written. However, the authors must address the following concerns.

1.     In the introduction, the author can talk about tryptophan and cortisol in relation to PTSD along with serotonin.

2.     Why only male participants were included? The lack of female participants is a major concern.

3.     Changes in serotonin, tryptophan and cortisol levels in PTSD patients have not been taken into account so far? How this work is different.

Author Response

Dear Editor,

In response to the review of the paper titled: ‘Changes in the Serum Concentration Levels of Serotonin,
Tryptophan and Cortisol among Stress-Resilient and
Stress-Susceptible Individuals after Experiencing Traumatic Stress’ (ijerph-2005992) the relevant changes suggested by the editor have been made.

In accordance with the recommendations of Reviewer 3:

    1. The introduction has been improved. The factors studied in relation to PTSD have been discussed
    2. By design, only men, working in hazardous conditions (miners, firefighters) and burdened with PTSD, participated in the study. This study did not assume the participation of women. I plan to study PTSD in women and men in another project in the future
    3. The novelty of this study is the comparison of PTSD-SR and PTSD-SS groups in terms of the parameters studied.

Kindest regards

The Author

Round 2

Reviewer 2 Report

Nothing